# Accuracy Tests and Precision Assessment of Localizing Underground Utilities Using GPR Detection

**DOI:** 10.3390/s21206765

**Published:** 2021-10-12

**Authors:** Krzysztof Ryszard Karsznia, Klaudia Onyszko, Sylwia Borkowska

**Affiliations:** 1Faculty of Civil Engineering and Geodesy, Institute of Geospatial Engineering and Geodesy, Military University of Technology (WAT), 00-908 Warszawa, Poland; sylwia.borkowska@wat.edu.pl; 2Department of Imagery Intelligence, Faculty of Civil Engineering and Geodesy, Military University of Technology (WAT), 00-908 Warszawa, Poland; klaudia.onyszko@wat.edu.pl

**Keywords:** ground penetrating radar, accuracy detection, soil moisture parameters, statistical testing

## Abstract

Applying georadar (GPR) technology for detecting underground utilities is an important element of the comprehensive assessment of the location and ground infrastructure status. These works are usually connected with the conducted investment processes or serialised inventory of underground fittings. The detection of infrastructure is also crucial in implementing the BIM technology, 3D cadastre, and planned network modernization works. GPR detection accuracy depends on the type of equipment used, the selected detection method, and external factors. The multitude of techniques used for localizing underground utilities and constantly growing accuracy demands resulting from the fact that it is often necessary to detect infrastructure under challenging conditions of dense urban development leads to the need to improve the existing technologies. The factor that motivated us to start research on assessing the precision and accuracy of ground penetrating radar detection was the need to ensure the appropriate accuracy, precision, and reliability of detecting underground utilities versus different methods and analyses. The results of the multi-variant GPR were subjected to statistical testing. Various analyses were also conducted, depending on the detection method and on the current soil parameters using a unique sensor probe. When planning detection routes, we took into account regular, established grids and tracked the trajectory of movement of the equipment using GNSS receivers (internal and external ones). Moreover, a specialist probe was used to evaluate the potential influence of the changing soil conditions on the obtained detection results. Our tests were conducted in a developed area for ten months. The results confirmed a strong correlation between the obtained accuracy and the measurement method used, while the correlation with the other factors discussed here was significantly weaker.

## 1. Introduction

Ground Penetrating Radar (GPR) detection technology is a recognised, classical method of detecting and locating hidden infrastructure. For many years, its advantages have been described in the extensive literature on the subject. Its theoretical basis, consisting of the emission and detection of electromagnetic waves of different modulation, has been presented in numerous publications [1,2,3,4,5,6]. A summary of this measurement technology development over the decade is presented in publication [7]. As its authors mention, the development concerns both instrumental issues (new solutions in the field of physics or software) and the technology of the measurement itself.

Georadars are used for various engineering tasks—from the inventory of underground infrastructure, the location of historical objects (underground passages, crypts, remains of foundations) to the detection of pipeline failures or leaks in tunnels [8,9,10]. The application of GPR technology may also encompass searching for tree root systems in intensively invested urban areas [11], karst forms in caves [12], archaeological remains [13], drainage systems [14], and many more. Apart from that, the discussed technology is applied in various geomorphological and geotechnical tasks. Currently, georadars are available in multiple forms of equipment used by specialists to pursue their goals. For example, broadband road georadars allow for advanced spatial detection of infrastructure that is hidden below the surface of roads [15].

On the other hand, mobile GPRs, integrated with other mapping instruments, such as laser scanners or photogrammetric cameras transported on platforms, enable us to obtain comprehensive 3D data of the measured space, both above- and underground. Small, manual construction georadars allow for the detection of installations located in the walls, covered by plaster, to improve the management of usable space in construction objects, also in terms of Building Information Modelling (BIM). Currently, such works are connected with virtual reality that enables the GPR operator to track the hidden infrastructure in real-time against the existing buildings or interior fittings [16].

Regardless of the equipment used, GPR detection consists of emitting electromagnetic waves and receiving and analysing the beam reflected from discontinuities in the structure of the analysed ground [17]. Such discontinuities may be caused either by objects located in soil or by layers of soil with significantly different properties [17].

Literature provides a variety of information about the obtained precision of GPR detection versus technical and environmental limitations. Technical data provided by certain georadar manufacturers [18,19,20] do not contain direct information about the average accuracy of location of the detected objects, usually being the utility infrastructure in the ground. However, they provide the technical data necessary to plan the detection of hidden infrastructure elements in a correct and optimised way. Moreover, national regulations and standards specify the requirements for maintaining records of technical utility networks [21,22]. As a result, when detecting hidden infrastructure, the operator should ensure the required accuracy and choose an adequate technology. Since detection results are influenced by various factors, such as the frequency of electromagnetic wave emission [23], the type of soils in which detection works are performed [24], as well as the material from which the hidden objects are made [25,26,27], one should be fully aware of them during the detection process. As a result, operators should use suitable equipment, ensure the repeatability of measurements, and, as far as possible, provide other sources of information that may complement the generated functional model.

To ensure high accuracy and reliability of the location of the detected utility networks, the work of georadars is supported by other instruments: mainly surveying, such as GNSS satellite receivers, which work in real-time and use differential adjustments [28] so that they offer the accuracy of 3D positioning ranging from one to several centimetres (usually 1 to 5 cm), as well as robotic electronic total stations [29]. The latter enables to conduct precise positioning works on a level lower than one centimetre [8]. The positioning of the georadar movement trajectory may also be supported by navigation GNSS receivers using different satellite systems that facilitate the work and improve the accuracy of positioning (e.g., SBAS) [28]. Such receivers are usually installed in the tablet that controls the operation of the device, and the precision offered by them, considering the additional system support mentioned before, ranges from several tens of centimetres to over one meter [8]. Apart from that, GPR detection should be supported with manual detectors of underground facilities employed in land surveying. Such an approach is recommended by manufacturers [30], entirely aware of the functional limitations of the application of georadars. These limitations include, among others, the dimensions of the equipment compared to the size of some detection areas (problems with penetrating locations that are difficult to access). Additionally, the accuracy of 2D positioning, compared to depth positioning, often becomes less satisfactory in light of legal regulation [21]. In such a situation, it becomes necessary to employ traditional positioning with the use of manual detectors and the tracking or galvanic methods [31], which offer the possibility to locate the traces of the course of specific underground utility networks.

The above circumstances pose a challenge for experts who use georadars in their professional activities and scientific research. The necessity to develop optimal methods to improve detection reliability becomes obvious, together with the need to verify the works in terms of their accuracy and repeatability. Such assessment should ideally be conducted with the use of statistical methods. Empirical sciences employ numerous types of statistical tests to verify the correlations between the relevant samples and populations. Such tests enable researchers to obtain plenty of additional information about the significance of the obtained measurement results. Depending on the type and characteristics of data, adequate tests can be used. However, it is sometimes challenging to select the most suitable test, as some are more suitable than others considering the actual characteristics of the analysed phenomenon. In such events, tests can be regarded as more or less “conservative”, meaning that they may not have considered the particular properties of specific datasets that are characteristic, for example, for the GPR measurements discussed here. Thus, it is necessary to select the statistical tests carefully.

What inspired us to analyse the topic of assessing the detection of underground infrastructure was not only the issue of evaluating the precision of positioning of various types of hidden utility networks. Our aim was also to assess the repeatability and reliability of works conducted using GPR in invested areas [32], and thus, the reliability of ground penetrating radar detection. To answer the question formulated in this way, we spent many months conducting georadar observations to support various positioning methods and subject the results to careful statistical tests. Another part of the works was performed with the use of a probe (“Penetrologger”) that determines the soil moisture content. By employing this surveying method, we aimed at finding potential correlations between various soil conditions with the GPR detection results considering statistical analyses. The research period included all seasons of the moderate climate (spring, summer, autumn, and winter), and the measurements were taken in conditions that are characteristic of the given season (the temperatures from +25 to −10 degrees Celsius and a thick snow cover). We aimed to perform a functional assessment of various detection methods and determine the most appropriate way to identify hidden underground infrastructure.

## 2. Materials and Methods

In our studies, we assessed the accuracy of GPR detection using the Leica DS2000 instrument. This device is equipped with a double-frequency antenna that enables simultaneous detection of objects located directly below the ground surface and, more profoundly, even several meters below ground [19]. The instrument employs two antennas offering different detection frequencies. The 700 MHz antenna detects objects with very high resolution but with a smaller depth range. On the other hand, the antenna with a lower frequency (250 MHz) has a lower resolution, but at the same time, it offers a more extensive depth range. Therefore, one can read the resolution and depth information from a given frequency. The resolution of GPR measurements is called the instrumental ability to determine a minimum distance at which two identical targets can be recognised as separate objects. One can distinguish vertical and horizontal resolutions—the increment values of changing antenna frequency and the dielectric constant of the medium. The vertical and horizontal resolution of the GPR method can be calculated from Formulas (1) and (2), respectively.
(1)$$∆d≅\frac{c}{4f\cdot \sqrt{\varepsilon_{r}}}$$
(2)$$∆a≅\sqrt{\frac{d\cdot c}{2f\cdot \sqrt{\varepsilon_{r}}}}$$
where:
c—velocity of propagation of an electromagnetic wave,f—antenna frequency,∆d—vertical resolution of the GPR method,∆a—horizontal resolution of the GPR method,$\varepsilon_{r}$—relative dielectric permittivity,d—depth of the reflection boundary.

Assuming that in the studied area there are mainly sandy soils with clay and silt which relative dielectric permittivity ranges from 11 to 36 and the antenna frequency is 250 MHz, the vertical resolution in the GPR method is 5–9 cm. For the 700 MHz antenna frequency, the vertical resolution is 2–3 cm. The horizontal resolution depends on the depth of the reflection boundary. As the depth of the located underground object increases, the value of the horizontal resolution shrinks. This resolution for the frequency of 250 MHz ranges from 2.2–3.3 cm (depth 0.5 m), 3.2–4.3 cm (depth 1 m) to 4.5–6 cm (depth 2 m). The horizontal resolution for the 700 MHz frequency it is equal to: 1.3–1.8 cm (depth 0.5 m), 1.9–2.5 cm (depth 1 m) and 2.7–3.6 cm (depth 2 m).

Such class of GPR equipment (DS 2000 by Leica Geosystems) is often used for map inventory works or design purposes, geophysical research, detecting objects in archaeology, etc. [1,33,34]. Some relevant technical data is presented in Table 1.

The view of the tested equipment is shown in Figure 1.

The expected accuracy of works may also be improved by additional encoders integrating the GPR device with external positioning systems, such as GNSS satellite receivers or robotic total stations (RTS). However, when choosing the technology reinforcing the tracking accuracy, one should consider the movement of the GPR (its speed may reach up to 10 km/h or even faster). It results in significant challenges for positioning, mainly due to the problems with obtaining “fix” solutions, i.e., capturing the total number of GNSS-wave phase cycles [29]. Similarly, when a georadar with an integrated GNSS receiver moves to the obscured areas, the visibility of available satellites will also be lost. Another factor leading to unstable positioning will be the multipath effect [35], which means that the GNSS signals are reflected from various terrain obstacles. Besides, one also should mention the necessity for real-time-network (RTN) access [36], improving the overall positioning accuracy.

The situation is similar while using a total station. In this case, the instrument “follows” the moving target placed on GPR and is marked by a surveying prism. Theoretically, such positioning allows the accuracy of ±1 cm affected by different error sources [37]. The functional assessment of using RTS may be found in numerous publications, e.g., [38]. Considering the above limitations, the authors decided to use only GNSS satellite methods augmenting GPR detection, apart from the traditional measurement with a defined detection grid.

Other environmental factors may also influence the GPR-detection accuracy, such as soil type [39] or water conditions [40]. Due to that, while experimenting, the authors decided to take additional measurements of soil moisture content using a specialist probe [41] shown in Figure 1c,d. The Penetrologger probing tool equipped with a Thetaprobe humidity sensor enables point-to-point measurement of soil moisture content. GPR-detection relies on the difference in the dielectric constant value [42] so, the soil should be analysed before GPR measurements. Based on the literature [43], the authors assumed that the soil moisture content would be another crucial parameter needing consideration.

The electromagnetic wave sent from the transmitting antenna of the georadar is subject to such phenomena like reflection, refraction, or attenuation while passing through geological media of different values of the dielectric constant. The dielectric constant, or the so-called relative dielectric permittivity $\varepsilon_{r}$, is expressed as the ratio of the dielectric permittivity of the material to that of a vacuum $\varepsilon_{0}$.
(3)$$\varepsilon_{r}=\frac{\varepsilon }{\varepsilon_{0}}$$

The higher the difference between the relative electric permittivity of the neighbouring media, the stronger is the reflection of the electromagnetic wave [44], which is exemplified in Table 2. The increase in the dielectric constant leads to a decrease in the speed of the electromagnetic wave.

Table 2 demonstrates that high contrast between two media usually occurs between air and any geological medium, water and ice, sand and clay, as well as peat and sand.

The determination of soil moisture content of the investigated soil in the GPR detection area using the probe mentioned above has become a valuable source of additional information that enriched our observation model.

### 2.1. Methodology of the Tests

The tests were conducted on the campus of the Military University of Technology in Warsaw, Poland. The geographic coordinates of the orientation centroid of the test area are φ = 52°15’16.786” N and λ = 20°54’11.325” E. Detailed visualization of the test site is available in the Polish National Geoportal at [44]. Apart from the situation, the portal also enables the visualization of the technical infrastructure network allowing the authors to optimise the GPR route planning (individual profiles were perpendicular to the course of underground installations).

The measurement site was an area of 55 × 80 m (Figure 2). Its choice was determined by the availability and diversity of technical infrastructure (low voltage power supply wires, heating, and water supply pipes). The analysis of available vector data revealed that there had been five technical infrastructure networks: low voltage power supply cable, gas network, telecommunication, and heating [45,46].

GPR detection was conducted in three ways: (1) using a regular grid aligned in the area, (2) with an internal integrated GNSS, and (3) an external receiver.

In the first method, detection measurements were executed along the rectangular grid with the baseline located parallel to the pavement (Figure 2), which resulted in 12 profiles. To assure appropriate georeferencing, we measured characteristic grid points using the GNSS Leica GS15 receiver. First, their cartographic coordinates were converted to the PL-2000 system (used in Poland) and saved in a .dxf file. Then, during the measurements, the baseline was drawn on the points imported from the receiver.

In the second method, the routes of characteristic profiles were automatically tracked using the integrated GNSS module integrated with the device controller without an independent external antenna (Figure 1b).

In the third method, the trajectory of GPR was assessed using a GNSS Leica GS15 receiver with an external antenna. The devices were paired wirelessly by Bluetooth utilizing the NMEA data exchange protocol used in satellite navigation [47].

In all three cases, the detection was carried out using two GPR antenna central frequencies: 250 MHz and 700 MHz. According to the technical specification, the maximum depth detection for the Leica DS2000 georadar is 200 ns, which, in theory, enables the detection of objects at a depth minor than 10 m. However, in our research, we have conducted our detection at a depth of approximately 5 m at 100 ns. The information about the scan interval allows the operator to set the distance between sending the impulses into the ground. In our case, this distance was approx. 4 cm. The strive dictated the selection of parameters to locate the course of both shallow and relatively deep underground objects optimally.

The three variants of GPR detection were augmented by additional soil moisture measurements taken with the Penetrologger probe. During the survey, relevant radargrams were generated, recorded, and processed with the uNext and GRED HD software (data post-processing and precise positioning of objects). Georadar images may be processed in 2D and 3D modes. Then, the detected subsurface objects can be vectorised. Figure 3 shows an example of the detection of underground utility networks with their position in the radargram. A fragment of the radargrams for each detected network concerns both 700 MHz and 250 MHz antenna frequencies.

The final stage of data processing relied on comparing the coordinates of characteristic points of the detected networks with their reference data obtained from the National Geodetic and Cartographic Resource [48]. The purpose of this comparison was to verify the manually selected points by reference-to-reference data.

The statistical tests carried out in this study were used to test the accuracy and repeatability of the detection of underground objects. To assess the accuracy and repeatability of the detection, the x and y coordinates reflecting the situational location of the network of utilities were used.

According to the literature [49,50], we can state that the authors always used a data set consisting of 3 and 5 measurement series, respectively, to test the repeatability of a GPR detection. Therefore, we also concluded that for our test, a similar data set would have been optimal. Thus, in our research, we obtained a number of 60 radargrams in five measurement series.

The stages of the processing of GPR data are presented in Figure 4. In the GRED HD software, the positioning and filtration system is automatically recognised, and the filter parameters are selected individually for the data set. Standard radargram processing, therefore, includes time-zero correction, removing the background, frequency, and gain filters. The program enables individual selection of parameters, which has been proven in publication [51].

For each type of utility network (divided into measurement series and depending on the selected measurement method), the standard error of the mean value ($M_{i,j,k}$) was calculated using Formula (4). As we performed several measurement cycles (depending on the method), the standard error of GPR detection $M_{j,k}$ (Formula (5)) according to the type of network and surveying method was based on unequally accurate observations, for which weights s$p_{i,j,k}$ were calculated (Formula (6)), where [pvv]=min. The results of calculating standard errors in the xy direction are presented in Table 3.
(4)$$M_{i,j,k}=\pm \sqrt{\frac{\left[vv\right]}{n\cdot \left(n-1\right)}}$$
(5)$$p_{i,j,k}=\frac{1}{M_{i,j,k}^{2}}$$
(6)$$M_{j,k}=\pm \sqrt{\frac{\left[pvv\right]}{\left[p\right]\cdot \left(n-1\right)}}$$
where:
i—measurement series number,j—measurement method number,k—the type of network (1–4),p—weights,[vv]—sum of squares of corrections,[p]—sum of weights,[pvv]—the product of the sum of weights and the sum of squares of corrections.

### 2.2. Statistical Testing

The authors subjected the obtained validation results to statistical testing with the use of parametric significance tests [52,53,54]. The primary assumption was the normal distribution of random variables, referred to as the Gauss distribution (its graphic representation will be the so-called Gauss Curve). In this case, 68% of the variables should appear close to the average value +/− standard deviation (uncertainty range). Apart from the mean and the standard deviation, the normal distribution is also characterised by skewness and kurtosis coefficients [55]. As the value of the skewness coefficient increases, the analysed distribution diverges from normal. A similar situation occurs for kurtosis or the flattening of the empirical distribution. The higher the value of kurtosis, the more the distribution is different from normal [56].

Our study analysed whether the detection results obtained in specific measurement periods for mononomial measurement methods belong to the same population. Additionally, we assessed whether the results obtained using different detection methods also belong to the same population. The final stage of our calculations consisted of determining the potential correlations between detection results and soil moisture content in specific measurement periods.

To determine whether the distribution of GPR detection results was normal, we performed two types of tests: the Shapiro-Wilk test [57], which was treated as the primary test, and the Kolmogorov-Smirnov [58]. Our choice was to compromise a broad set of similar statistics and their practical applications in land surveying [59]. According to the literature [60,61], the Shapiro-Wilk test is the best method to verify the normality of the data distribution, considering that the datasets are small, with potential outliers. As a result, we obtain the so-called W statistic. The higher the W-parameter, the closer the empirical distribution is to normal. This test may be described with Formula (7):(7)$$W=\frac{{\left[\sum_{i}a_{i}\left(n\right)(X_{n-i+1}-X_{i})\right]}^{2}}{\sum_{j=1}^{n}\sum {\left(X_{j}-\bar{X}\right)}^{2}}$$
where:
W—the result of the Shapiro-Wilk test,$a_{i}\left(n\right)$—constant, values in the W distribution table,$X_{n-i+1}-X_{i}$—difference between extreme observations,j—subsequent observations in the given sample,i—subsequent differences between extreme observations,$\bar{X}$—average value.

On the other hand, the Kolmogorov-Smirnov test works better with larger sets of more homogeneous data [56,58].

The compatibility of populations was analysed using the Fisher-Snedecor test (Test F) [62]. This statistic allows checking the equality of variance in two populations characterised by a normal distribution [63]. Test F is used to analyse the significance of differences for two variances. It is based on the assumption that the given populations have a normal distribution $N\left(m_{1},\sigma_{1}\right)$ and $N\left(m_{2},\sigma_{2}\right)$ and that the population of samples is, respectively, $n_{1}$ and $n_{2}.$ Then, the value of the Fisher-Snedecor test parameter is calculated with the Formula (8).
(8)$$F=\frac{s_{1}{}^{2}}{s_{2}{}^{2}}$$
where:
F—the value of the Fisher-Snedecor test parameter,$s_{1},s_{2}$—values of standard deviations for series of results, assuming that $s_{1}>s_{2}$.

The test statistic was the basis for determining the *p*-value, which is compared with the level of significance α. If the *p*-value was lower than the level of significance α, the null hypothesis would show that the variances in both groups were equal and should be rejected. In the opposite case, there were no grounds for rejecting the null hypothesis. The *p*-value refers to the probability of making an error in accepting the hypothesis that differences between mean values exist. It should be added that the same test is used, among others, to validate the accuracy of surveying instruments according to ISO standards 17123 [64].

Our analyses were also complemented with the tests of potential correlation between the obtained detection results and the soil moisture measurement results. To collect such data we utilised a special sensor called “Penetrologger”. The literature provides many applications for a soil surface moisture probe with the evaluation of soil products [65,66]. The sensor measures the volumetric moisture of the soil with an accuracy of 1% [67]. It consists of four 6 cm-long steel rods forming a system, which impedance is dependent on the moisture content of the soil [68].

The dependence between two variables may be determined using an indicator referred to as the correlation. For example, it may be determined by applying the Bravais-Pearson correlation that determines the linear correlation between random variables or Spearman correlation (for monotonic dependence of data) [69].

Our assumption was verified by analysing the Bravais-Pearson linear correlation based on the assumption that the dependence between the detection results and soil moisture content was linear. The Bravais-Pearson linear correlation coefficient can be determined using the Formula (9):(9)$$r_{xy}=\frac{\sum \left(X_{i}-\bar{X}\right)\cdot \left(Y_{i}-\bar{Y}\right)}{\sqrt{\sum {\left(X_{i}-\bar{X}\right)}^{2}\cdot \sum {\left(Y_{i}-\bar{Y}\right)}^{2}}}=\frac{\frac{1}{n}\sum X_{i}Y_{i}-\bar{X}\bar{Y}}{\sigma_{X}\sigma_{Y}}=\frac{cov\left(X,Y\right)}{\sigma_{X}\sigma_{Y}}$$
where:
$X_{i}$, $Y_{i}$—i-the valuers of observation from populations *X* and *Y*,$\bar{X},\bar{Y}$—means from populations *X* and *Y*,$\sigma_{X},\sigma_{Y}$—standard deviations of populations *X* and *Y*,n—number of observations (*X* and *Y* have the same number of observations).

The higher the absolute value of the *r_xy_* coefficient, the stronger is the existing correlation between random variables [56]. In our studies, we performed all calculations using the Statistica v13.3 software package [70].

## 3. Results

### Results of Statistical Tests

Table 4 presents the results of the Shapiro-Wilk test that verifies the normal distribution of data. The value of the W statistics is in each case lower than the critical value, which proves that the obtained distribution of specific detection results is normal. Tests were conducted for the significance level *α* = 0.05.

The results of the Kolmogorov-Smirnov test (for specific utility networks and depending on the GPR detection method) are presented in Table 5.

It is worth noting the results of the KS test for the heating network (for the second method). The obtained value of 0.05 suggests that the hypothesis saying that the distribution is normal should be rejected. However, considering that the SW test was used as the primary test, the authors have concluded that this hypothesis should not be rejected due to the relatively small size of the tested dataset and the strength of the SW test [58].

The normal distribution for specific measurement periods (1–5) was also analysed with the Shapiro-Wilk and Kolmogorov-Smirnov tests. A visual assessment of the course of normal distribution concerning the accuracy of detection of technical infrastructure was also performed, which is presented in histograms (Figure 5). One may notice that there are no grounds for rejecting the null hypothesis for any of the samples. It proves that the conducted measurements are characterised by the normal distribution.

To demonstrate the repeatability of detection of the relevant utility networks performed in various seasons (measurement series), let us analyse the results of the Fisher-Snedecor test. Based on Table 6, we can conclude that the hypothesis stating that variances in the corresponding populations are equal was rejected in five cases (for method 1) and in one case (for methods 2 and 3), respectively.

As mentioned before, the GPR detection was supported each time by the measurement of soil moisture content. The measurement points were evenly distributed over the entire study area. The measurement of the humidity of a total of 40 points (in each measurement series) was carried out at a depth of 15 cm. Figure 6 shows the maps of isohumes generated in Surfer v13.3 software [71] for individual test periods.

The Bravais-Pearson correlation coefficient may be used to determine the linear correlation between the accuracy of detection of the given utility network and the interpolated value of soil moisture.

The measurement data was divided into five periods (detection conducted in June, August, September, November 2020, and January 2021). The soil moisture content in the test area in specific measurement periods was, respectively: 8–31% (first measurement period), 4–25% (second measurement period), 3–21% (third measurement period), 7–25% (the fourth measurement period), and 12–27% (the fifth measurement period). These values fluctuate slightly between different seasons of the year. The visual assessment of the correlation between GPR detection accuracy (in direction x and y) and soil moisture for specific measurement periods is presented in Figure 7.

The maps of isohumes reveal that in summer and autumn, soil moisture values in the tested area were slightly lower than in winter and spring. Figure 7c shows that the third test period was the only one where the prerequisite for considering a slight correlation between GPR detection accuracy and soil moisture occurred. In this case, the correlation coefficient was 0.3. This leads to the conclusion that GPR detection conducted in various seasons of the year is equally accurate, and the influence of soil moisture on its accuracy may be omitted.

## 4. Discussion

The presented experiments and the corresponding results point out the usefulness of the proposed analyses. The research consisted of real data sets. The proposed data processing methodology allowed for selecting the most appropriate method of capturing echograms. Based on the statistical tests, one can conclude that the most optimal approach is employing a georeferenced baseline and the rectangular grid during measurements.

The available literature tells us a lot about the statistical testing in the accuracy and repeatability assessment of non-invasive detection of subsurface objects [72]. Based on the available publications, it is known that the correlation between the depth of the object and soil moisture is significant [73,74]. Our article attempted to determine the correlation between the accuracy of GPR detection in the XY direction and soil moisture. The research on the potential impact of soil moisture on the accuracy of GPR detection showed a slight correlation in the studied area. Undoubtedly, the conducted analyses could be enriched with additional measurement series; however, one of the main challenges in empirical studies, especially those related to the field surveying, is finding an optimal testing method versus the expected accuracy. This approach (tested in various external conditions) would allow for a reliable correlation assessment between these two factors.

## 5. Conclusions

GPR detection is usually conducted in various external conditions that may generate errors and modify the level of the obtained measurement uncertainty. Although the technical documentation provided by georadar manufacturers with the equipment contains the necessary technical data and specifies the expected accuracy in proper measurement conditions, it usually refers to uniform values. During important field works that require high accuracy and reliability, it is recommended to perform nominal validation tests of the equipment used. The authors have proposed a comprehensive method of assessing the accuracy and reliability of GPR detection. The research included the analysis of three detection methods, with detection performed in all seasons under the influence of changing external conditions.

Additionally, the authors conducted soil moisture measurements in the analysed area to determine the potential influence of these values on the accuracy of GPR detection. Field tests were compared with reference data from the National Geodetic and Cartographic Resource and then subjected to comprehensive statistical evaluation. Our research revealed that the average accuracy of GPR detection was (respectively, for measurement methods 1, 2, and 3): 0.03 m, 0.05 m, 0.06 m for the power network, 0.03 m, 0.04 m, and 0.07 m for the gas network, 0.02 m, 0.08 m, and 0.04 m for the heating network, and, finally, 0.04 m, 0.03 m, and 0.01 m for the telecommunications network. The measurement series from the homogenous test area confirm the repeatability of the occurrence of the detected networks. In most of the obtained radargrams, all the existing technical infrastructure was detected. However, the lowest percentage of detected subsurface objects referred to the heating network.

Furthermore, the correlation between soil moisture content in the analysed area and the accuracy of infrastructure detection was practically non-existent. Regarding that, this greatly facilitates the reliable detection of the said device in typical areas. It concludes that obtaining the highest GPR detection accuracy demands several measurement cycles, knowing that the positioning would be the least reliable when using an integrated GNSS receiver. Such cases certainly apply to locations with limited horizon visibility and terrain obstacles. In situations like this, the most effective method would be the traditional grid stake out in the field using known surveying techniques.

## Figures and Tables

**Figure 1 sensors-21-06765-f001:**
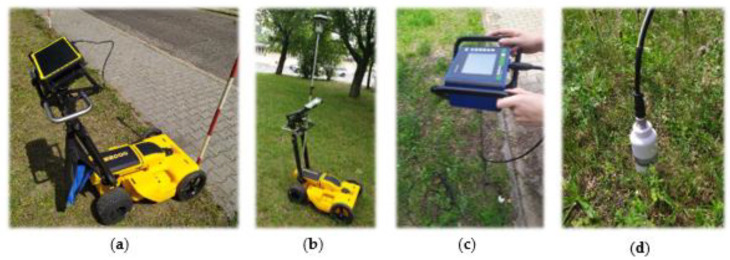
View of the Leica DS2000 GPR used during the infrastructure detection test: (**a**) general view; (**b**) GNSS receiver integrated with GPR; (**c**) Eijkelkamp Penetrologger sensor; (**d**) view of the soil sensor.

**Figure 2 sensors-21-06765-f002:**
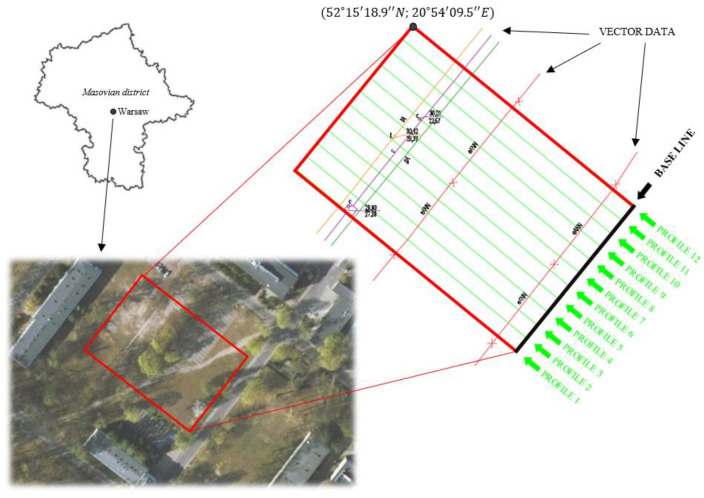
Test area with baseline and marked measurement profiles.

**Figure 3 sensors-21-06765-f003:**
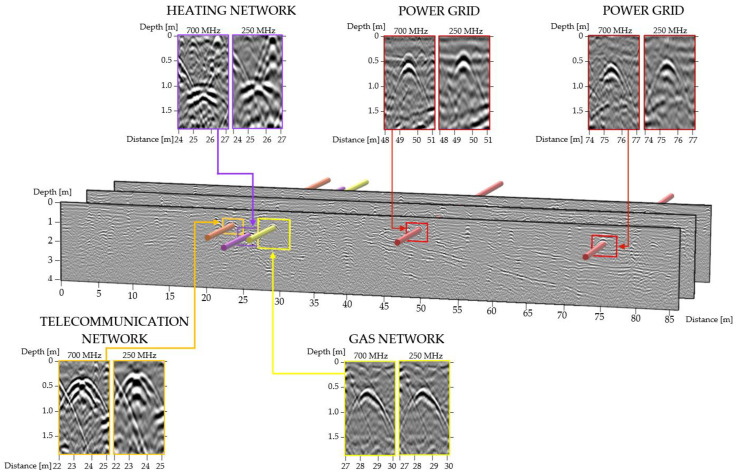
Schematic view of the detected underground infrastructure in front of the radargram.

**Figure 4 sensors-21-06765-f004:**
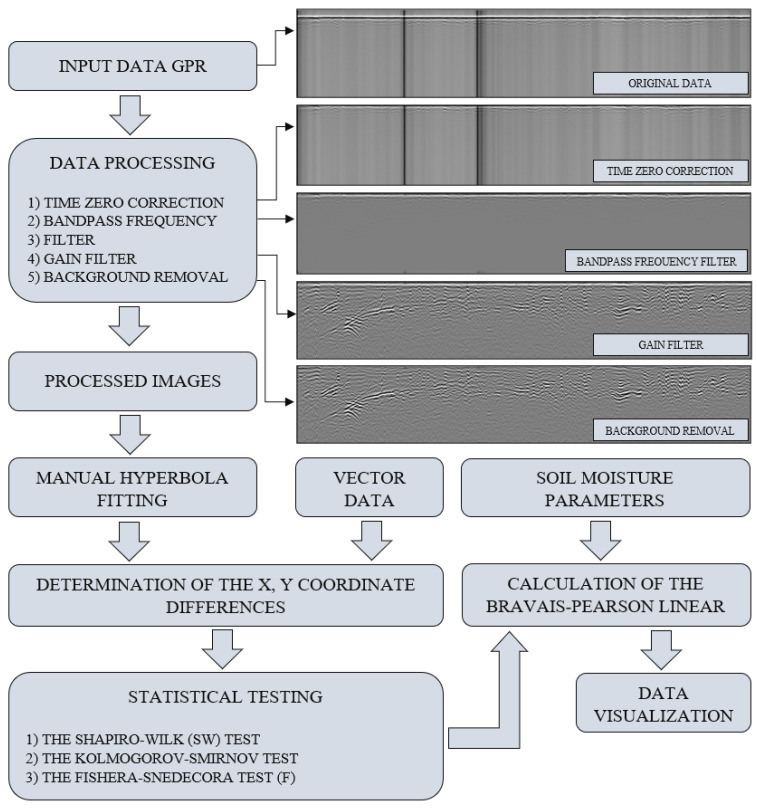
Diagram showing the stages of research work.

**Figure 5 sensors-21-06765-f005:**
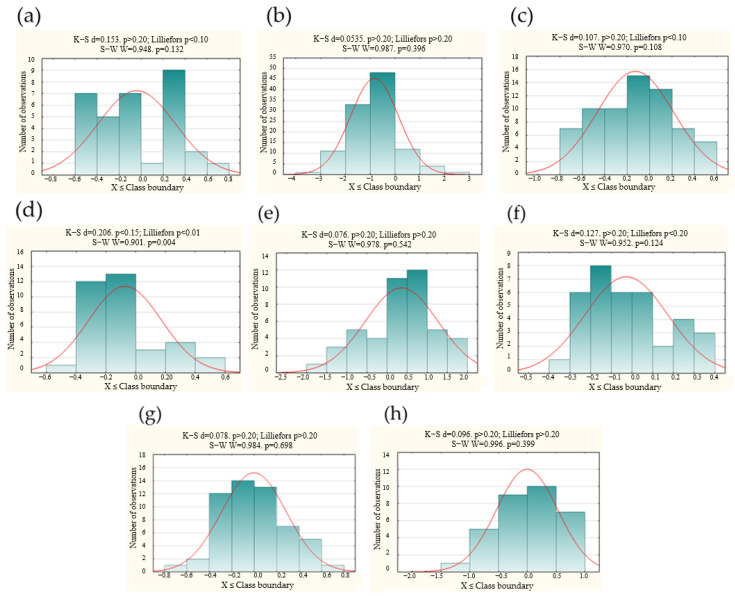
Histograms presenting the normal distribution of the accuracy of GPR detection (concerning three survey methods) for specific test periods: (**a**,**b**) the 1st measurement period; (**c**) the 2nd measurement period; (**d**,**e**) the 3rd measurement period; (**f**,**g**) the 4th measurement period; (**h**) the 5th measurement period. Histograms (**a**,**c**,**d**,**f**) refer to method 1, (**b**,**e**,**h**) to method 2, and (**g**)—to method 3.

**Figure 6 sensors-21-06765-f006:**
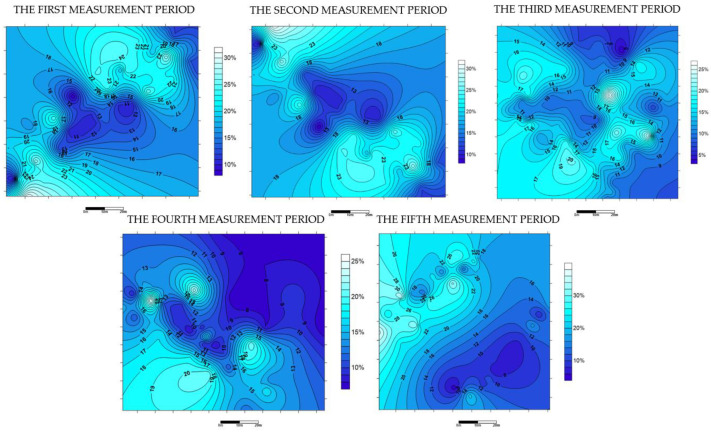
Maps of the soil moisture in the test area for different measurement periods.

**Figure 7 sensors-21-06765-f007:**
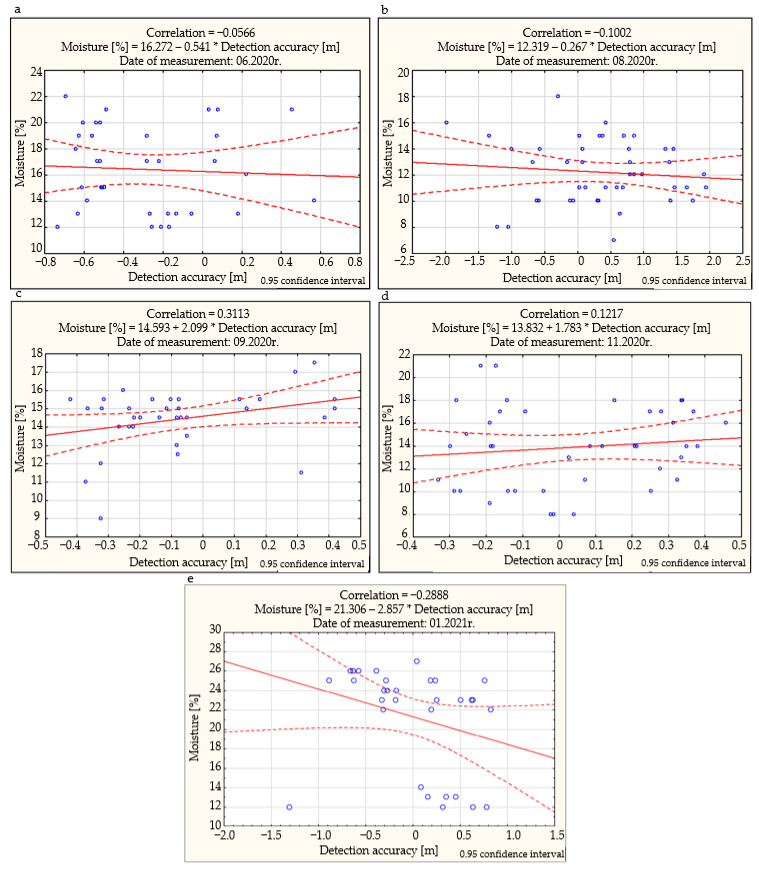
Results of the determination of the Bravais-Pearson coefficient in 5 measurement periods: (**a**) 06.2020, (**b**) 08.2020, (**c**) 09.2020, (**d**) 11.2020, (**e**) 01.2021.

**Table 1 sensors-21-06765-t001:** Technical data about the DS 2000 Leica GPR.

Feature	Specifications
antenna footprint	0.4 m–0.5 m
antenna frequencies	250 MHz 700 MHz
sampling frequency	400 kHz
scan rate per channel	381 scans/s
scan interval	42 scans/m

**Table 2 sensors-21-06765-t002:** Relative electric permittivity depending on the medium [44].

Medium	$\varepsilon_{r}$
air	1
water	81
asphalt	2.5–3.5
concrete	3–9
ice	3.2
snow	1.4
dry sand	3–5
sand saturated with water	20–30
sandy soil	11–18
silt	14–36
clay	25–36
limestone	6–11
peat	50–78

**Table 3 sensors-21-06765-t003:** Standard errors of georadar detection obtained for three measurement methods.

The Type of Network	The Average Error of GPR Detection [m]		
	Method 1.	Method 2.	Method 3.
Power grid	0.03	0.05	0.06
Gas network	0.03	0.04	0.07
Heating network	0.02	0.08	0.04
Telecommunication network	0.04	0.03	0.01

**Table 4 sensors-21-06765-t004:** Table showing the results of the SW.

Type of Technical Infrastructure	Shapiro-Wilk Test (SW)					
	1. Method		2. Method		3. Method	
	*W*	*W_kr_*	*W*	*W_kr_*	*W*	*W_kr_*
power grid	0.982	0.960	0.977	0.957	0.972	0.918
gas network	0.973	0.927	0.980	0.940	0.939	0.850
heating network	0.954	0.940	0.961	0.934	0.906	0.850
telecommunication network	0.972	0.927	0.977	0.934	0.930	0.818

**Table 5 sensors-21-06765-t005:** Table showing the results of the KS.

Type of Technical Infrastructure	Kołmogorov-Smirnov Test (SW)					
	1. Method		2. Method		3. Method	
	*p*	*α*	*p*	*α*	*p*	*α*
power grid	0.20	0.05	0.10	0.05	0.20	0.05
gas network	0.20	0.05	0.20	0.05	0.20	0.05
heating network	0.15	0.05	0.05	0.05	0.10	0.05
telecommunication network	0.20	0.05	0.20	0.05	0.20	0.05

**Table 6 sensors-21-06765-t006:** Table showing the results of the F.

The First Method				The Second Method				Third Method			
Measurement Series		*p*	Hypothesis	Measurement Series		*p*	Hypothesis	Measurement Series		*p*	Hypothesis
power grid				power grid				power grid			rejected hypothesis
1	2	0.03	rejected hypothesis	1	2	0.72	no reason to reject the null hypothesis	1	2	0.03	
5	2	0.04		1	3	0.61		gas network			no reason to reject the null hypothesis
2	4	0.04		1	4	0.66		1	2	0.35	
1	3	0.06	no reason to reject the null hypothesis	2	3	0.44		heating network			
1	4	0.93		2	4	0.92		1	2	0.35	
2	3	0.70		3	4	0.40		telecommunication network			
3	4	0.08		5	3	0.08		1	2	0.89	
5	1	0.96		5	1	0.02	rejected hypothesis				
5	3	0.08		5	2	0.01					
5	4	0.98		5	4	0.01					
gas network				gas network							
1	2	0.87	no reason to reject the null hypothesis	1	2	0.62	no reason to reject the null hypothesis				
1	3	0.84		1	3	0.39					
1	4	0.64		1	4	0.62					
2	3	0.97		2	3	0.82					
2	4	0.71		2	4	0.99					
3	4	0.74		3	4	0.82					
5	1	0.73		5	1	0.81					
5	2	0.83		5	2	0.52					
5	3	0.86		5	3	0.44					
5	4	0.86		5	4	0.80					
heating network				heating network			rejected hypothesis				
1	2	0.54	no reason to reject the null hypothesis	1	2	0.04					
1	3	0.11		1	3	0.06	no reason to reject the null hypothesis				
1	4	0.47		1	4	0.54					
2	3	0.32		2	3	0.59					
2	4	0.86		2	4	0.19					
3	4	0.48		3	4	0.34					
5	1	0.90		5	1	0.18					
5	2	0.68		5	2	0.64					
5	3	0.20		5	3	1.00					
5	4	0.59		5	4	0.46					
telecommunication network				telecommunication network							
1	3	0.02	rejected hypothesis	1	2	0.57	no reason to reject the null hypothesis				
1	4	0.01		1	3	0.22					
1	2	0.07	no reason to reject the null hypothesis	1	4	0.23					
2	3	0.30		2	3	0.47					
2	4	0.10		2	4	0.49					
3	4	0.77		3	4	0.99					
5	1	0.06		5	1	0.03	rejected hypothesis				
5	2	0.81		5	2	0.01					
5	3	0.45		5	3	0.01					
5	4	0.22		5	4	0.01					

## Data Availability

The data presented in this study is available on request from the co-author Klaudia Onyszko, e-mail: klaudia.onyszko@wat.edu.pl.

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
