# Peer review of "Accuracy Tests and Precision Assessment of Localizing Underground Utilities Using GPR Detection"

_sensors, 2021, doi:10.3390/s21206765_

Round 1

Reviewer 1 Report

Comments

Section 1

Numbering of references must be successive throughout the manuscript. Now it is not (see paragraph 70 for negative example).

Section 2 Materials and methods

The vertical and horizontal resolution can be calculated in different ways, explain why did these formulas were used (1)(2)?

Information about soil type is twice mentioned in the text - paragraph 142 „in the studied area there are mainly sandy soils with clay and silt [43]˝ and the paragraph 203-204 ˝The dominant types of 204 soil in the analyzed area are sandy soils with clay and silt [43].˝ There is no need for that.

Table 2 is mentioned in the paragraph 143 and then in the paragraph 154 is mentioned Table 1. Numbering of tables must be successive throughout the manuscript.

Section 3.1

In the paragraph 230 add letter “b” to the Fig. 1.

In the paragraphs 257-259 it is stated that “The final stage of data processing relied on comparing the coordinates of characteristic points of the detected networks with their reference data obtained from the National Geodetic and Cartographic Resource [46].”

There is a need for better explanation of performed statistical tests.

What coordinates of characteristic points were compared and statistically tested (maybe x, y or z)?

For example in the paragraph 275 it is written that the “the standard error of the mean value (??,?,?) was calculated using formula 4”.

It has not been explained which coordinate values were used in calculation of mean value. Maybe the distance in one or all of the x, y or z direction errors were calculated.

Formulas (4) and (6) are not for calculating the standard error of the mean value.

Please explain what is [vv], [pvv], [p] and pi,j,k.

Table 3 is representing standard errors of georadar detection in [m]. Please explain in which direction is that error, x, y or z?

Section 4.1

There are one Table 2 and two Tables 3. These numbers of tables have been already taken in other sections.

Figure 7 has detection accuracy. Please explain in which direction is that accuracy measured, x, y or z?

Best regards

Author Response

Answers to the reviewers

----------------------------------------------------------------------------------------------------------------------------------------

Reviewer 1#

Dear Reviewer,

Thank you very much for your valuable remarks and suggestions. We have done our best to improve our manuscript respectively. We have marked all relevant changes driven by your requests in yellow colour. We consider your hints very helpful as well as we believe that the submitted material looks significantly better now.

Below, please find the answers (A) to your precious questions and comments (Q):

Q: Numbering of references must be successive throughout the manuscript. Now it is not (see paragraph 70 for negative example).

A: Yes, we agree with you. We have corrected our text, respectively.

Q: The vertical and horizontal resolution can be calculated in different ways; explain why did these formulas were used (1)(2)?

A: Several formulas for calculating the resolution of GPR measurements can be found in the available literature, for example: [Rial F. I. et al., 2009*]. Knowing the values of the relative dielectric permittivity, we have chosen one of the available methods.

*Rial F. I., Pereira M., Lorenzo H., Arias P., Novo A. Resolution of GPR bowtie antennas: An experimental approach. Journal of Applied Geophysics 2009, Volume 67, Issue 4, Pages 367-373.

Q: Information about soil type is twice mentioned in the text - paragraph 142 "in the studied area there are mainly sandy soils with clay and silt [43]˝ and the paragraph 203-204 ˝The dominant types of 204 soil in the analyzed area are sandy soils with clay and silt [43].˝ There is no need for that.

A: Yes, we agree with you. We removed the repeated passage from the text.

Q: Table 2 is mentioned in the paragraph 143 and then in the paragraph 154 is mentioned Table 1. Numbering of tables must be successive throughout the manuscript.

A: Yes, we agree with you. We corrected this in our text.

Q: In the paragraph 230 add letter "b" to the Fig. 1.

A: Yes, we have added the letter "b" to Fig. 1 (sub-chapter 2.1 Methodology of the tests).

Q: In the paragraphs 257-259 it is stated that "The final stage of data processing relied on comparing the coordinates of characteristic points of the detected networks with their reference data obtained from the National Geodetic and Cartographic Resource [46]." There is a need for better explanation of performed statistical tests.

A: We added an explanation in the text. (marked with yellow in the sub-chapter 2.1 Methodology of the tests)

Q: What coordinates of characteristic points were compared and statistically tested (maybe x, y or z)? For example in the paragraph 275 it is written that the "the standard error of the mean value (??,?,?) was calculated using formula 4". It has not been explained which coordinate values were used in calculation of mean value. Maybe the distance in one or all of the x, y or z direction errors were calculated.

A: Yes, we did not write it in the text. The x and y coordinates reflecting the location of the utility networks were used to assess the accuracy and repeatability of the detection itself. We have included this information in the text (marked with yellow in the sub-chapter 2.1 Methodology of the tests).

Q: Formulas (4) and (6) are not for calculating the standard error of the mean value.

A: We have added a relevant explanation in our text. According to the known state-of-the-art for land surveying, also for the statistics, a mean error of a singular observable is expressed by the equation:

where (n - k) is a degree of freedom – in our case (n -1). The standard error of an average value is expressed as:

where n is the number of observables.

Hence, after mathematical transformation:

Considering that each observable has its weight, which is usually defined as:

we obtain the ultimate formula:

The principle of error transition is commonly used in land surveying and can be found in numerous handbooks, for example, in Ghilani C. D. and Wolf P. R., Adjustment Computations: Spatial Data Analysis, Fourth Edition, John Wiley & Sons, Inc., ISBN: 978-0-471-69728-2, 2006 but also in many other publications. Sometimes the formula is given with different symbols, but it generally belongs to the surveying fundamentals.

Q: Please explain what is [vv], [pvv], [p] and pi,j,k.

A: We have added a relevant explanation in the text (marked yellow in the sub-chapter 2.1 Methodology of the tests). In general:

  • [vv] is a sum of squares of corrections,
  • [pvv] is a sum of weighted squares of corrections,
  • [p] is a sum of weights,
  • pi,j,k is a weight of a particular observable (i,j,k – series).

Such notification is typical for land surveying and belongs to the adjustment fundamentals.

Q: Table 3 is representing standard errors of georadar detection in [m]. Please explain in which direction is that error, x, y or z?

A: The results of calculating standard errors in the XY direction are presented in Table 3. We added an explanation in the text (marked with yellow in the sub-chapter 2.1 Methodology of the tests).

Q: There are one Table 2 and two Tables 3. These numbers of tables have been already taken in other sections.

A: Yes, we do agree with you. We have corrected this item in our text.

Q: Figure 7 has detection accuracy. Please explain in which direction is that accuracy measured, x, y or z?

A: We have added relevant information about the accuracy direction. The visual assessment of the correlation between GPR detection accuracy (in direction x and y) and soil moisture for specific measurement periods is presented in Figure 7 (marked with yellow in the sub-chapter 3.1. Results of statistical tests).

We wish to thank you again for your helpful review and remain with

Kind regards,

The authors

Reviewer 2 Report

Review of the manuscript sensors-1356785 “Accuracy tests and precision assessment of localizing underground utilities using GPR detection” by Krzysztof Ryszard Karsznia, Klaudia Onyszko and Sylwia Borkowska.

This is a paper dealing with the accuracy of the location of underground utilities by means of GPR data. The authors obtain several GPR profiles at different periods of the year as well as direct measurements to determine the soil moisture conditions. A statistical analysis of the locations of the utilities with different positioning methods along the year is done. Although the manuscript has potential interest for the readers, I think that it is not properly organized, and the interpretations are not supported by the data. Consequently, I consider that it cannot be published in its present form. More details are given below:

- After reading the manuscript, I am not able to understand if the accuracy test are related only to the horizontal positioning of the utilities or to the vertical (depth) location too. If only the horizontal location is considered, the work has not enough scientific sounding, as it is clear that the horizontal location is hardly related to the soil moisture content. If the accuracy results obtained comprise both the horizontal and depth locations of the utilities, two different sources of information are being mixed, because the depth location is largely associated to the moisture content that modifies the soil electric permittivity. 

Consequently, I have many doubts about the meaning of the accuracy results obtained in the manuscript. For example, the values of horizontal accuracy largely depends on the velocity of the antenna during the data acquisition. The faster the velocity the lower the accuracy, but there is not any description about how the velocity of the data acquisition was maintained at a constant rate for the 12 profiles and for all the time periods (i.e. the 60 profiles). On the other hand, as the accuracy is of a few cm, it is a key point that the antenna was positioned always at the beginning of the profiles with a precision  of less than 1 cm, and this is not clear at all if it has been done and how. As a result, I consider that it is not possible to compare the errors of the three methods at the 5 different measurement periods of the year if they are not obtained exactly at the same velocity and with exactly the same starting point. It is also noteworthy that the accuracy errors obtained by the authors range from 1 to 7 cm, that fall in the same range that the positioning errors of the GNSS systems used (1 to 5 cm, line 77). This means that it is not possible to discriminate if the accuracy errors are due to the GNSS system, to the influence of the soil moisture content, or to any other factor such as the antenna velocity. Moreover, crucial information about the electric permittivity value used to convert the TWT values into real depth is totally missing. Taking into account the large range of permittivity values described by the authors for the soil of the study area (Table 2), this data is of crucial importance to evaluate the accuracy of the vertical location. 

- Another major problem of the manuscript is the lack of a Discussion section where the results from this study were compared with similar previous works. It is necessary to find similar works in the literature to be compared to, in order to state the new findings of this works, the improvements in relation to previous studies as well as the limitations found. The discussion must not repeat the main findings but it must describe the pros and cons of the proposed method.

Apart from these major considerations, more specific comments re below:
-    At lines 143-151, the authors provide horizontal and vertical resolution values for 250, 400 and 700 Mhz frequency antenna, but they only used 250 and 700 Mhz antennae so the values corresponding to 400 Mhz make no sense here.
-    At table 1 it is said that the sampling frequency of the DS 2000 Leica GPR is 400 Mhz what is a major mistake, because the sampling frequency of the equipment is 400 kHz.
-    The table showing the results of the Fisher-Snedecor test only compares the measurements series 1 with 2 for the third method. These means that, whereas for methods 1 and 2 the profiles were made at 5 different measurement series, for the third positioning method only two different measurements series were carried out. Why? If the measurement series are different for the three positioning methods, this invalidate the statistical meaning of the accuracy tests.
-    Table 2 is cited before Table 1.
-    Table 2 (line 364 is really table 4). Table 3 (line 372 is really table 5). Table 4 (line 382 is really table 6).
-    The maps of soil moisture content (figure 6) need a color bar to be interpreted. On the other way, nothing is said about the location of the measurement points used to obtain the maps. How many measurements points were obtained and where were they located at the study area? At what depth where the measurements done?

Author Response

Dear Reviewer,

Thank you very much for your valuable remarks and suggestions. We have eagerly followed your recommendations and corrected our text, respectively. While doing that, we have marked all relevant changes in yellow.

Below, please find the answers (A) to your precious questions and comments (Q):

Q: After reading the manuscript, I am not able to understand if the accuracy test are related only to the horizontal positioning of the utilities or to the vertical (depth) location too. If only the horizontal location is considered, the work has not enough scientific sounding, as it is clear that the horizontal location is hardly related to the soil moisture content. If the accuracy results obtained comprise both the horizontal and depth locations of the utilities, two different sources of information are being mixed, because the depth location is largely associated to the moisture content that modifies the soil electric permittivity.

A: We intended to analyze the overall accuracy (and performance) of the GPR used to detect underground utilities using adjustment computation methodology. We are entirely aware that such a holistic approach concerns both horizontal as well as vertical accuracy. Nevertheless, both cases require factual investigation, which – by following our proposed policy, would be very difficult to be presented in one limited paper. Hence, we decided to split them and investigate separately. Here, we have focused on the horizontal aspect, which is affected by the tracking method (using classical stakeout network or GNSS-aided tracking) and the detecting performance of the device used. We assumed that the soil moisture might have influenced the detection accuracy. Regarding that, we have performed our test. We want to indicate that using Penetrologger was considered as an additional method, not the leading one. What concerns the vertical detection, in our profession, its accuracy may be essential for 3D-cadastral issues, and hence, its investigation requires thorough studies on – among others – relevant legal demands and regulations. Regarding that, such as approach seems to be extensive and requires focusing primarily on this item, which is the subject of our other current studies.

Q: Consequently, I have many doubts about the meaning of the accuracy results obtained in the manuscript. For example, the values of horizontal accuracy largely depends on the velocity of the antenna during the data acquisition. The faster the velocity the lower the accuracy, but there is not any description about how the velocity of the data acquisition was maintained at a constant rate for the 12 profiles and for all the time periods (i.e. the 60 profiles).

A: In our test, we used a standard velocity of a marching man doing the detection. We did not examine the GPR in extreme conditions and followed the manufacturer's suggestions and technical requirements. We wanted to explore the device in a standard way used in land surveying – like ISO-testing etc. The summoned ISO standards' primary assumption is to traditionally examine instrument accuracy by reducing possible error sources, which might affect final results. So, was our intention too.

Q: On the other hand, as the accuracy is a few cm, it is a key point that the antenna was positioned always at the beginning of the profiles with a precision  of less than 1 cm, and this is not clear at all if it has been done and how. As a result, I consider that it is not possible to compare the errors of the three methods at the 5 different measurement periods of the year if they are not obtained exactly at the same velocity and with exactly the same starting point. It is also noteworthy that the accuracy errors obtained by the authors range from 1 to 7 cm, that fall in the same range that the positioning errors of the GNSS systems used (1 to 5 cm, line 77). This means that it is not possible to discriminate if the accuracy errors are due to the GNSS system, to the influence of the soil moisture content, or to any other factor such as the antenna velocity. Moreover, crucial information about the electric permittivity value used to convert the TWT values into real depth is totally missing. Taking into account the large range of permittivity values described by the authors for the soil of the study area (Table 2), this data is of crucial importance to evaluate the accuracy of the vertical location.

A: As aforementioned, we wanted to examine the instrument accuracy by using the statistical testing standard for land surveying. Please note that the manufacturer's mounting set of GNSS antenna is fixed, and the GPR was tested similarly to other, different GPR-aided works. What is more, the track of moving GPR covered the entire terrain so that each utility was detected repeatedly (back and forth). Please note that we had known the location of each utility before we started – regarding that, we were able to test the instrument performance reliably. Even if the accuracy of GNSS were poorer than expected, we would have been able to localize to the whole path of a particular cable because it was crossed many times. Regarding that, one source of error was significantly limited. What is more, we did not rely only on GNSS. Instead of that, we had rather focused on a classical grid stakeout with a classical surveying method (orthogonal surveying with the mean accuracy of ±5cm). Such precision is entirely acceptable in land surveying as well as in the requirements of the utility cadastre. Finally, the vertical accuracy assessment is the subject of our current work. Based on that, we will remember your suggestions while preparing another paper.

Q: Another major problem of the manuscript is the lack of a Discussion section where the results from this study were compared with similar previous works. It is necessary to find similar works in the literature to be compared to, in order to state the new findings of this works, the improvements in relation to previous studies as well as the limitations found. The discussion must not repeat the main findings but it must describe the pros and cons of the proposed method.

A: Yes, we agree with you. We have added a relevant discussion section to the chapter Discussion.

Q: At lines 143-151, the authors provide horizontal and vertical resolution values for 250, 400 and 700 Mhz frequency antenna, but they only used 250 and 700 Mhz antennae so the values corresponding to 400 Mhz make no sense here.

A: Yes, we agree with you. We corrected it in our text.

Q: At table 1 it is said that the sampling frequency of the DS 2000 Leica GPR is 400 Mhz what is a major mistake, because the sampling frequency of the equipment is 400 kHz.

A: Yes, we agree with you. We corrected this in our text in table 1.

Q: The table showing the results of the Fisher-Snedecor test only compares the measurements series 1 with 2 for the third method. These means that, whereas for methods 1 and 2 the profiles were made at 5 different measurement series, for the third positioning method only two different measurements series were carried out. Why? If the measurement series are different for the three positioning methods, this invalidate the statistical meaning of the accuracy tests.

A: Thank you for your question. In our tests, we have rather focused on the relevant surveying methods. The terrain was the same, as well as the observer and the surveying art. The device accuracy was comparable, and the data processing – also identical. Objective factors caused the lack of two cycles. Despite that, we decided to compare each series, assuming that such a comparison will be relevant, considering the overall accuracy suggested by the manufacturer.

Q: Table 2 is cited before Table 1.

A: Thank you for your remark. We have corrected this error respectively.

Q: Table 2 (line 364 is really table 4). Table 3 (line 372 is really table 5). Table 4 (line 382 is really table 6).

A: Yes, we agree with you. We corrected it in our text.

Q: The maps of soil moisture content (figure 6) need a colour bar to be interpreted. On the other way, nothing is said about the location of the measurement points used to obtain the maps. How many measurements points were obtained and where were they located at the study area? At what depth where the measurements done?

A: Yes, we agree with you. We added a colour bar to our maps of soil moisture. Additionally, we have added some relevant information about soil moisture measurement to our text. The measurement points were evenly distributed over the entire study area. The soil moisture measurement of a total of 40 points (in each measurement series) was carried out at a depth of 15 cm (marked with yellow in the sub-chapter 3.1. Results of statistical tests). Finally, to assure the best cartographic quality, we have improved our maps by changing their colour scale (now following the principles of visualizing humidity features), correcting the fonts etc.

We wish to thank you again for your helpful review and remain with

Kind regards,

The authors

Round 2

Reviewer 2 Report

The revised version has been improved and many suggestions have taken into account. Altough I still have many doubts about the scientific soundness of the results because of the issues previously exposed about the control of the antenna velocity and the accuracy of the starting measuring point for all the experiments, resulting in an uncertainty that is assumed to be related to the positioning system, I agree that the manuscript can be published in its present form.

This manuscript is a resubmission of an earlier submission. The following is a list of the peer review reports and author responses from that submission.

Round 1

Reviewer 1 Report

Dear Authors,
Your article is well structured and clearly presented. One minor suggestion to improve the layout is that the tables could take up a little less space.

Reviewer 2 Report

The Authors propose a study to investigate the performance of GPR in localizing underground utility.

Generally speaking, the work essentially appears as an advanced exercise. The type of application is well known and well established. The step forwards indicated by this work is not substantial or at least not enough significant.

I would suggest considering the following two works, it is really important to elaborate more the novelty to be considered for publication

Pettinelli, Elena, et al. "GPR response from buried pipes: Measurement on field site and tomographic reconstructions." IEEE Transactions on Geoscience and Remote Sensing 47.8 (2009): 2639-2645.

Soldovieri, Francesco, et al. "GPR estimation of the geometrical features of buried metallic targets in testing conditions." Progress In Electromagnetics Research B 49 (2013): 339-362.

The most significant part is connected to the study of the impact of the soil moisture. It is not clear how the approach can perform when no accurate information are available. Also, in some specific condition, and for large depths, information could be not available.

Reviewer 3 Report

This paper addresses an interesting topic dealing with the accuracy of pipe detection with GPR. However, this reviewer consider that major improvement is needed before acceptable for publication. My recommendations is therefore to reject and encourage authors to resubmit. 

My main criticisms are: 

  • Page 2, line 53: The meaning of the acronym BIM is Building Information Modelling.
  • Page 6, lines 219-220: authors state "The maximum depth detection for the Leica DS2000 georadar is 200 ns". I am disagree with this sentence. The penetration depth of the signal depends on the dielectric properties of media. What media characterization are authors assuming to ensure a maximum depth of aprox. 10 m? 
  • The correct name for a GPR image is radargram, not echogram. 
  • In Figure 3, the reflection obtained for the Telecommunication networks seems to be two reflections overlaped? are you sure it is a unique pipe? 
  • Regarding the methodology to estimate the average error. In my opinion, authors should consider also the uncertainty, since they are providing unknown information. Some aspects should be also consider when dealing with GPR accuracy; for example, the resolution of the antenna frequency used. Authors should provide the resolution of the frequencies used in their work. 
  • With respect to the GPS, how authors guarantee the stable coupling of the GPS antenna to the GPR antenna? What was the correlation between the position of the GPR signal and the position of the GPS signal? 
  • Page 9, lines 292-293, authors state "the kolmogorov-Smirnov test works better with larger sets of more homogeneous data". More explanation is expected. Moreover, how large is your dataset? 
  • Figure 5 is illegible. 
  • Is your dataset enough to analyze repeatability of detection? How many data is usually employed from the published literature? Moreover, is your scenary representative enough?
  • Figure 7 is illegible. 
  • Page 15, line 396. Authors mention a Figure 8c, but there is not a Figure 8 in the paper. 

Reviewer 4 Report

Dear Authors, I believe that your contribution would be interesting and suitable for Sensors but it presents some “dark” parts for geophysicists.

1)    FIG.3 You used a double frequency GPR (250 – 700 MHz). What antenna do you show? (I suppose 250 MHz). I think it is interesting to see the dual antenna on the same target. Different frequencies have different penetration and resolution. The radargrams don’t have vertical/horizontal scales! See the two radargrams with power grid: we have a polarity change. Why?

2)    219-221 lines. You speak in general about GPR depth. Usually in the soil (clay, wet send, ….) the electromagnetic velocity is 6-7 cm/ns while you consider 10 cm/ns. The more realistic depth for 200 ns is << 10 m.

3)    All your paper is focused on “accuracy - precision”. You never talk about the GPR resolution that depends on the wave velocity and the frequency. I think you must write about resolution. (Velocity can be calculated with best fitting method). Then, what is the dimension of your targets?  

4)    Fig. 4 Data processing can greatly change the amplitude and phase signal. For this reason I think is important a figure “raw data vs processed data”. Then the parameters are important. You use Bandpass frequency (range?) and Filters (what?), Gain (AGC? Spherical divergence? Absorption?). Background removal is to be avoided, it can introduce ghosts and signal deformations.

5)    Bibliography. There are several references in Polish [3, 6,…]. It’s not easy for an ordinary reader follow them. Can you change them into some equivalent English reference?

6)    In GPR echogram is not used, radargram is used. MegaHertz is written MHz.

7)    Abstract could be improved by focusing on the method used in the paper.

I hope that you will find the comments to be of use to you and am looking forward with interest to read your final work.

Good work 

LB